# Unleashing the Potential of LLMs for Quantum Computing: A Study in Quantum Architecture Design

## Abstract

Large Language Models (LLMs) contribute significantly to the development of conversational AI and have great potential to assist scientific research in various areas. This paper attempts to address the following questions: What opportunities do the current generation of generative pre-trained transformers (GPTs) offer for the development of noisy intermediate-scale quantum (NISQ) technologies? Additionally, what potential does the forthcoming generation of GPTs possess to push the frontier of research in fault-tolerant quantum computing (FTQC)? In this paper, we implement a Quantum GPT-Guided Architecture Search (QGAS) model, which can rapidly propose promising ansatz architectures and evaluate them with application benchmarks including quantum chemistry and quantum finance tasks. Our results demonstrate that after a limited number of prompt guidelines and iterations, we can obtain a high-performance ansatz that is able to produce comparable results that are achieved by state-of-the-art quantum architecture search methods. This study provides a simple overview of GPT's capabilities in supporting quantum computing research while highlighting the limitations of the current GPT at the same time. Additionally, we discuss futuristic applications for LLM in quantum research.

## 1 Introduction

Large-scale language models (LLMs), such as ChatGPT (OpenAI, 2023), can learn autonomously from data and communicate with humans by reinforcement learning with human feedback (RLHF) mechanisms (Christiano et al., 2017). Since the release of ChatGPT by OpenAI, this type of AI technology has made a huge impact on a large number of research areas (van Dis et al., 2023; Stokel-Walker & Van Noorden, 2023). LLMs are being rapidly adopted in areas such as advanced chemistry (Pan, 2023), healthcare (Korngiebel & Mooney, 2021), protein design (Ferruz & Höcker, 2022), etc. However, how can LLMs help the emerging fields such as quantum computing? We have seen many recent articles discussing how quantum computing can play an important role in the development of generative pre-trained transformers (GPTs) in the form of exploiting more complex, larger search spaces and more efficient training (Baker, 2023), but no one has yet discussed whether GPTs can contribute to the development of quantum computing.

As the field of quantum computing evolves, the design of quantum architecture stands as one of the most critical aspects. Quantum architectures define the structure and behavior of quantum circuits, which underpin every quantum algorithm. The correct selection and optimization of quantum circuits are crucial to achieving superior computational speed-ups and minimizing error rates inherent in quantum systems. As such, the development of effective methodologies for quantum architecture design is a pressing need.

The design of quantum architecture is a complex task that requires a deep understanding of quantum mechanics (Chen et al., 2022; Cheng et al., 2020; Mato et al., 2022; Baker et al., 2021; Qi et al., 2023; Liang et al., 2023; Zhan & Gupta, 2022; Hillery et al., 2023; Liang et al., 2022b), computer science (Wang et al., 2022b; Hu et al., 2022; Li et al., 2023; Wang et al., 2021; 2022c), and optimization algorithms (Baheri et al., 2022; He et al., 2023). Traditional methods have relied heavily on human expertise and intuition, which, while invaluable, are inherently limited by our cognitive

capabilities and biases. This is where large language models (LLMs) like GPT can make a significant contribution. GPT models are renowned for their ability to understand, generate, and reason about human language. They can absorb vast amounts of information, identify patterns within it, and make inferences based on these patterns. These capabilities make them particularly suited for the task of quantum architecture design. In the context of quantum architecture design, GPT can serve in multiple capacities. For instance, in tasks like Variational Quantum Eigensolvers (VQE), LLMs can act as controllers to conduct classical optimization. LLMs can effectively navigate the vast and complex search spaces of classical parameters, potentially outperforming traditional optimization techniques.

Furthermore, LLMs can be leveraged to explore the design space of the ansatz circuit. They can generate efficient architectures for the ansatz based on patterns and principles gleaned from large datasets of quantum circuits and their performance metrics. By doing so, they can contribute to the development of more expressive and efficient quantum architectures.

In summary, the application of GPT and similar models to quantum architecture design represents a promising avenue for accelerating the progress of quantum computing. By integrating human expertise with the power of advanced machine learning, we stand to make significant strides in our quest to harness the full potential of quantum computing. In this paper, we aim to delve deeper into this intriguing prospect and explore how GPT can contribute to the design of quantum architectures.

## 2 RELATED WORKS

**Ansatz Architecture Search:**    The design of ansatz architectures plays a crucial role in the research of VQAs. A well-crafted ansatz architecture can lead to more accurate computational results (Kandala et al., 2017). In the past, researchers derived problem-specific ansatz structures by analyzing particular problems (Peruzzo et al., 2014; O'Malley et al., 2016). For instance, the Unitary Coupled-Cluster Singles and Doubles (UCCSD) scheme (Grimsley et al., 2019) is still considered the "golden" ansatz for solving molecular energy problems within the VQE framework.

QuantumNAS (Wang et al., 2022a) and QAS (Du et al., 2022) introduced noise-aware frameworks for the automated search of ansatz architectures. On the other hand, PAN (Liang et al., 2022a) and Layer VQE (Liu et al., 2022) proposed ansatz frameworks that employ progressive training and search approaches at both pulse and gate levels, respectively. These advancements have expanded the range of possibilities for designing ansatz structures and optimizing their performance in various quantum computing applications. In contrast to conventional search strategies, we employ a method that iteratively prompts GPT-4 to propose ansatz architecture from a given search space, this kind of paradigm that combines human feedback and the ability of GPT-4 can greatly reduce the cost for search ansatz structure.

**Powerful Capabilities of LLMs:**    The remarkable performance of Large Language Models (LLMs), such as GPT, has left a huge impression on researchers. These models have demonstrated significant assistance or indispensable acceleration and resource-saving contributions in various research domains (Brown et al., 2020; Ouyang et al., 2022; van Dis et al., 2023). We have observed that in closely related research directions, researchers have begun to appreciate and utilize LLMs to generate datasets (Ye et al., 2022) and accomplish sentence embedding tasks more effectively (Cheng et al., 2023). In the field of computer architecture, there have been efforts to use GPT for neural architecture search, and the effectiveness of GPT has been demonstrated through several simple examples (Zheng et al., 2023; Yan et al., 2023). Beyond computer science, researchers in health care (Korngiebel & Mooney, 2021), advanced chemistry (Pan, 2023; Bran et al., 2023), and other domains (Ferruz & Höcker, 2022) have explored the integration of GPT with existing execution tools to further advance the research process in their respective fields. Our work aims to explore the roles and limitations of GPT in the emerging field of quantum computing, thereby contributing to the understanding of the potential and challenges in applying these powerful models to this exciting new domain.

## 3 UNDERSTANDING THE DESIGN OF QUANTUM ARCHITECTURE FOR VQAS

The architecture of the ansatz circuits in Variational Quantum Algorithms (VQAs) is pivotal, as it underpins the ability of these algorithms to tackle complex computational tasks effectively and

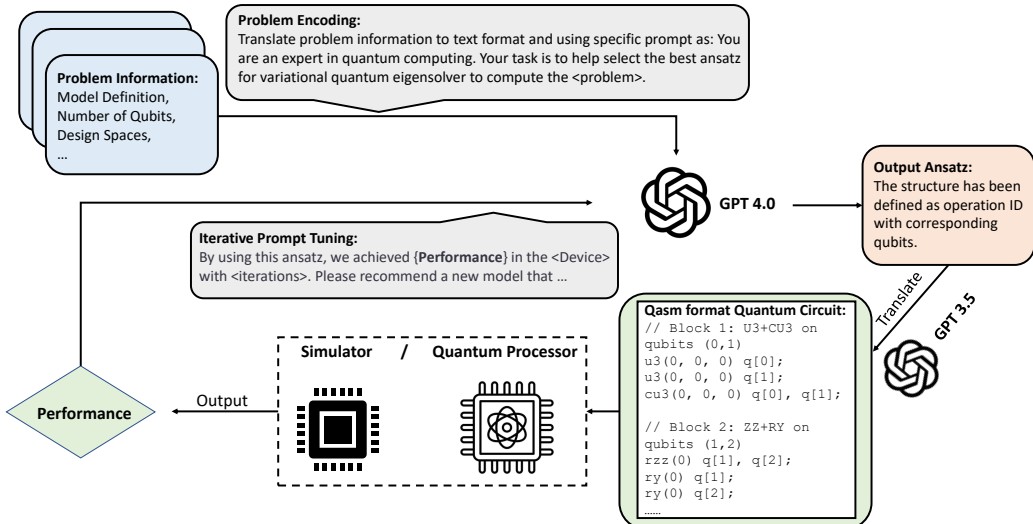

Figure 1: Illustration of design and implementation of QGAS. Following an initial encoding issue, GPT-4 suggests an ansatz, which is subsequently processed by a sub-model constructed by GPT-3.5 to convert it into QASM format. To assess the effectiveness of the ansatz, a benchmarking application is executed, enabling the evaluation of its quality. The obtained results are then fed back to GPT-4 through a natural language prompt, facilitating further iterations and refinements.

efficiently. The ansatz circuit's task is to approximate the lowest energy quantum state, commonly known as the ground state. A direct correlation exists between the performance of the VQA and the ansatz circuit's proficiency in emulating this quantum state.

Specifically, the Hamiltonian associated with the target molecule is converted into a sequence of Pauli matrices, thereby transforming the continuous problem of identifying the molecular ground state energy into a discrete optimization task. This transformation is a core operation in quantum computing, demonstrating the potential of VQAs for quantum supremacy, particularly in situations where classical computers encounter significant computational hurdles. The architecture of the ansatz circuit is deeply entrenched in the principles of quantum entanglement and superposition. Quantum gates, the primary building blocks of ansatz circuits, are strategically arranged to manipulate qubits, the fundamental units of quantum information. These gates facilitate unitary transformations, driving the evolution of the quantum system from an initial state to a final state that closely approximates the quantum system's ground state.

The design of the ansatz circuit often adheres to heuristic principles and is highly tailored to the specific problem being addressed. Factors such as the selection and sequencing of quantum gates, as well as the degree of qubit entanglement, are critical elements that require customization. An effectively designed ansatz circuit should allow efficient exploration of the Hilbert space, the vector space encompassing all conceivable states of the quantum system. Researchers have proposed various strategies for designing ansatz circuits, such as:

1. Hardware-efficient ansatz (Liang et al., 2022a; Kandala et al., 2017): These are circuits designed to be compatible with specific quantum hardware, considering the connectivity and native gate set of the quantum processor.

2. Problem-specific ansatz (Peruzzo et al., 2014; O'Malley et al., 2016): This approach incorporates prior knowledge of the problem at hand (e.g., the structure or symmetries of the target molecule) to design the ansatz circuit more efficiently.

3. Machine learning-guided ansatz (Wang et al., 2022a; Rattew et al., 2019): Machine learning techniques are employed to generate and optimize ansatz circuits by learning from previously solved instances or based on heuristics.

In conclusion, the architecture design of the ansatz circuit is a key factor determining the performance of VQAs. Researchers continue to explore novel ways to improve the design and optimization of ansatz circuits to maximize the potential of VQAs for practical applications.

## 4 METHODOLOGY

In the search for quantum circuit architectures, GPT-4 is capable of recommending ansatz structures under the guidance of prompts iteratively. Variational Quantum Algorithms (VQAs) represent a quantum computing approach that utilizes hybrid quantum-classical algorithms to solve optimization problems and simulate quantum systems (Peruzzo et al., 2014). The choice of ansatz is of critical importance in VQAs as it determines the efficiency and accuracy of the algorithm. In this section, we introduce our proposed QGAS model and its training process as showin in the Fig. 1. The core objective of QGAS is to leverage GPT as a controller to recommend high-quality ansatz structures for VQAs.

### 4.1 ANSATZ ARCHITECTURE GENERATION

We initially provide GPT-4 with a description of the corresponding problem and our requirements, ensuring the granularity of the description is as detailed as possible to obtain reliable responses from GPT-4. For instance, in the case of quantum chemistry problems related to molecular ground state energy, we provide information about the molecule, the basis for fitting molecular electron orbitals, and the required number of qubits. For quantum finance investment optimization problems, we supply information regarding the budget, the number of assets, the risk factor, and stock details.

In our preliminary model, we present GPT with six design spaces (Wang et al., 2022a; McClean et al., 2018; Lloyd et al., 2020; Farhi & Neven, 2018; Henderson et al., 2020; McKay et al., 2018), corresponding quantum circuit interfaces, and code:

(1) `U3+CU3` – One block has a U3 layer with one U3 gate on each qubit and a CU3 layer.

(2) `ZZ+RY` – One block contains one layer of ZZ gate and one RY layer.

(3) `RXYZ` – One block has four layers: RX, RY, RZ, and CZ.

(4) `ZX+XX` – Based on their MNIST circuit design, one block has two layers: ZX and XX.

(5) `RXYZ+U1+CU3` – Based on their random circuit basis gate set, we propose a design space in which one block has six layers in the order of RX, RY, RZ, CZ, U1, and CU3.

(6) `IBMQ Basis` – One block with the basis gate set of IBMQ devices, in which one block has six layers in the order of RZ, X, RZ, SX, RZ, and CNOT.

Simultaneously, we grant GPT-4 the choice of qubit placement of the design spaces. The default number of circuit blocks is set to six, and each circuit block takes two qubits, e.g., the output should be an ID list for the ansatz as well as the selected qubits for each block. For example: [1, (0,1)], [2, (1,2)],..., [0, (4,5)] means we use operation1 for block1 and the block1 is on qubits(0,1), operation2 for block2 and block2 is on qubits (1,2),..., operation 0 for block6 and the block6 is on qubits (4,5). Subsequently, we request GPT-3.5 to output the recommended ansatz structure in QASM format since the resource for GPT-4 is limited for regular users, some sub-tasks can be efficiently done by GPT-3.5 to save the cost on GPT-4.

### 4.2 TRAINING ON THE GENERATED ANSATZ

Utilizing the generated quantum circuit structures, we train an ansatz within the context of the corresponding problem to achieve enhanced performance. Considering different problems with distinct characteristics, we obtain the Hamiltonians corresponding to these problems using various methods. For instance, in quantum chemistry for molecular ground state energy, we perform a fit of the electronic structure of the molecule and subsequently obtain the Hamiltonian of the related molecule through fermionic mapping (Anselmetti et al., 2021; Uvarov et al., 2020). In contrast, for general quantum finance problems, we encode the problem into a quadratic problem structure and then translate this quadratic problem into an Ising Hamiltonian (Barkoutsos et al., 2020; Camino et al., 2023).

---

**Algorithm 1** GPT-Prompts ($l_{des}$, $l_{perf}$, *Design Space*, *Choices*)

---

// Input: a list of explored design $l_{des}$, the corresponding normalized performance of each design $l_{perf}$, design backbone *Model*, and design space *Choices*.

// Output: a prompt to the GPT-4 model;

$prompt_s$ = "*You are an expert in the field of quantum computing, especially for quantum architecture design.*";

$prompt_u$ = "*Your task is to help select the best ansatz for variational quantum eigensolver to compute the ground state energy of [name] molecule. The ansatz works on [number] qubits and contains [number] blocks. For each block, there are 6 types of operations to choose from:*" {*Design Space*} *Please output an ID list for the ansatz as well as the selected qubits for each block.* {*Choices*};

*For example: [1, (0,1)], [2, (1,2)], ...., [0,(4,5)] means we use operation 1 for block1 and the block1 is on qubits(0,1), operation2 for block2 and block2 is on qubits(1,2), ...,operation 0 for block6 and the block6 is on qubits(4,5).*"

// Input the [Choices] to GPT-3.5 model;

// Translate [Choices] to QASM form;

**return** $prompt_s + prompt_u$;

---

Following this, we execute the relevant problem with the proposed ansatz using a quantum processor or simulator. Generally, in gate-level quantum circuit training, we employ gradient-based training methods, and the optimization process relies on calculating the gradients of the parameterized quantum circuits with respect to their parameters. These gradients are utilized to update the parameters in order to minimize the target objective function, such as the difference between the calculated and target energy values.

## 4.3 Human Feedback & Capability of GPT

Quantum architectures are crucial for quantum computing applications. The selection of an appropriate ansatz can significantly impact the computational cost and the accuracy of results. In this section, we investigate the cooperation between human expertise and the GPT model to provide a more efficient structure for the ansatz, a framework we refer to as QGAS.

Previous results on quantum architecture search, such as the state-of-the-art framework Quantum-NAS (Wang et al., 2022a), often proceed in two steps. The first step involves searching for a super-circuit within a large predefined design space. The second step involves searching for a subcircuit within this supercircuit. In this process, every design space within this area is treated with equal importance when sampled. The QGAS approach also begins with a search within a large predefined design space. However, it differentiates by incorporating human feedback and specific design space characteristics to perform a ranking. The ranked design spaces are then used to guide the second step of the search for the ansatz. In the context of the QGAS framework, each design space is treated differently based on its unique features such as expressivity, entanglement power, and others. With the assistance of the GPT model, more granular supervision and consideration are performed within the same predefined space. Initially, experts in quantum computing bring their deep understanding of quantum systems' complex behavior to bear, providing feedback on specific search strategies. They advise the search algorithm to focus on particular parameters or structures that have shown promise in past research, leveraging established quantum theories and past experiences. This guidance helps eliminate redundant exploration of the search space, improving the search's efficiency and accuracy. The GPT-4 model also contributes to this process by suggesting additional strategies for quantum circuit optimization. For instance, it may recommend leveraging initial parameters for a "warm start" or incorporating certain error mitigation methods. However, it is crucial to remember that the suggestions from the GPT-4 model can be a mixture of effective and incorrect strategies. Here, human expertise is necessary to discern the effective methods from the incorrect ones, preventing wastage of resources on the latter.

Furthermore, human feedback is paramount in evaluating the search outcomes. The final design of a quantum architecture must be theoretically feasible and practically superior in performance. Even if a design scores highly in the algorithm, it cannot be accepted if it fails to pass the rigorous evaluation by human experts. These evaluations consider various factors such as theoretical consis-

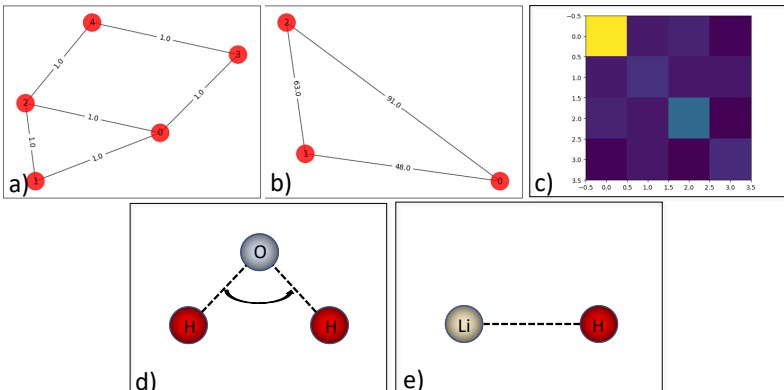

Figure 2: Visualization of benchmark applications. a) Max-Cut problem with 5 nodes. b) Traveling Salesman Problem with 3 nodes. c) Portfolio Optimization problem. d) Molecule structure of $H_2O$. e) Molecule Structure of $LiH$.

tency, practical viability, and comparison with existing architectures. This feedback then feeds back into the system to update and adjust the search strategies, establishing an iterative feedback loop. The integration of human feedback and GPT's power thus enables a more efficient and effective exploration of the quantum circuit architecture. This is especially beneficial for complex quantum computations where the optimal circuit design is not intuitively obvious, and the number of possible circuit configurations can be prohibitively large.

## 5 EXPERIMENTS

### 5.1 EXPERIMENT SETUP

Our experiment relies on GPT-4 as the selected LLM, and we both use Torchquantum (Wang et al., 2022a) and qiskit-runtime (Johnson, 2022) to execute experiments. As for the experiments on qiskit-runtime, we import the system model from $ibmq\_manila$.

### 5.2 APPLICATION BENCHMARKING FOR QGAS

In order to verify the efficacy of the proposed QGAS framework, we employed a series of application benchmarks. These benchmarks have been chosen to represent diverse domains of quantum computing, showcasing the versatility and broad applicability of the QGAS framework. Fig. 2 illustrates five distinct applications, namely: Portfolio Optimization, MaxCut Problem, Traveling Salesman Problem (TSP), and the estimation of Molecule Ground State Energy for both Lithium Hydride (LiH) and Water ($H_2O$).

The Portfolio Optimization application originates from the sphere of quantum finance. This problem involves the selection of a set of investments that provides the greatest expected return for a given level of risk. By leveraging the strengths of quantum computing, we can optimize complex financial portfolios in quantum algorithms. And we set the risk factor as 0.5, the number of assets as four, the budget as half of the number of assets, and the penalty as equal to the number of assets, the visualization of the problem is shown in the Fig. 2(c). The MaxCut problem and the TSP, on the other hand, are examples of quantum optimization problems. MaxCut is a well-known problem in graph theory, and it involves the partitioning of a graph into two subsets to maximize the sum of the weights of the edges crossing the subsets, we generate a five nodes graph as shown in the Fig. 2(a). The TSP is a classic algorithmic problem focused on optimization, where it is necessary to find the shortest possible route that visits a set of locations and returns to the origin, we generate a graph with three destinations for the traveling salesman as shown in the Fig. 2(b), to be noticed, this problem will request 8 qubits to solve. Lastly, the estimation of Molecule Ground State Energy for LiH and $H_2O$ is a fundamental problem in quantum chemistry, the structure of these two molecules has shown in Fig. 2(d) and (e). The ability to precisely calculate the ground state energy of molecules is key for understanding chemical reactions and designing new molecules and materials.

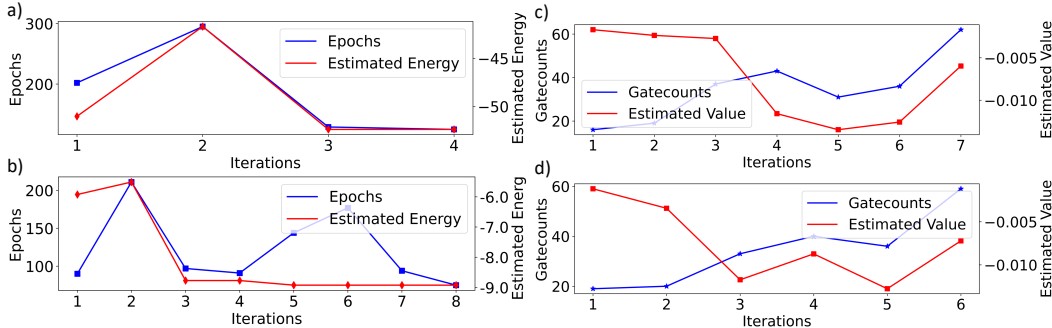

Figure 3: (a) and (b) Experiments for two trials of the quantum chemistry molecule ground state energy tasks and evaluation of the ansatz architecture generated by QGAS. We show both the epochs and estimated energy for each iteration, where a lower estimated energy with fewer epochs indicates a better ansatz. (c) and (d) Experiments for two trials of the portfolio optimization problem and evaluation of the ansatz architecture generated by QGAS. We show both the gatecounts and estimated value for each iteration, a lower estimated value with smaller gatecounts indicates a better ansatz. In both trials, the results demonstrate in a limited number of prompt guidelines and iterations, we can obtain a high-performance ansatz generated by QGAS.

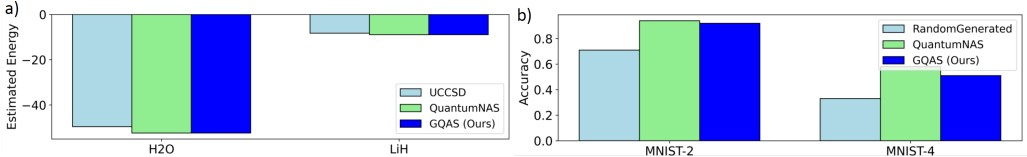

Figure 4: The application benchmarks for comparing the state-of-art ansatz and QGAS-generated ansatz. a) Molecule ground state energy estimation tasks for $H_2O$ and $LiH$, compare the ansatz generated by QGAS, UCCSD, and QuantumNAS. b) Machine learning tasks for MNIST-2 and MNIST-4 classification, compare the QGAS-generated ansatz, random generated ansatz, and QuantumNAS.

The optimization and finance problems are firstly encoded to quadratic problems and then mapped to Ising Hamiltonian. For the chemistry tasks, the molecules were first subjected to STO-3G approximation to model the electronic orbitals. Subsequently, a fermionic mapping was employed to derive the corresponding Hamiltonian.

## 5.3 EVALUATION AND COMPARISON

We evaluated our QGAS framework on both TorchQuantum and a simulator with the noise and system model of IBMQ Belem. The results were extensively compared with different existing ansatzes and state-of-the-art ansatz architecture search methods.

From Table 1, we can gain multiple observations, in the case of the Portfolio Optimization problem, the ansatz architecture found by QGAS outperformed the TwoLocal and RealAmplitudes architectures. However, the gate count was seven more than that of TwoLocal. Despite this, we believe the trade-off is justifiable considering the performance gain. The performance of the ansatz architectures and corresponding estimated gate counts at different iterations during model optimization is presented in Fig. 3 (c) and (d). It can be observed that during the optimization iterations, QGAS actively improves the quantum circuit's generalizability by increasing gate counts, thereby improving performance. And for both two trails, we obtain high-performance ansatz in a limited number of iterations. Moreover, we observe that when the gate counts reach a threshold, the performance then cuts down. We validated this gate count threshold for the portfolio optimization problem in our experiments with RealAmplitudes and TwoLocal as well. In the 8-qubit Traveling Salesman Problem, we observed a similar advantage that QGAS has over the other two ansatzes. For the 5-qubit Max-Cut problem, RealAmplitudes produced the best results, using eight more gates than the ansatz architecture found by QGAS.

Table 1: Comparison between existing ansatz and the ansatz generated by QGAS.

(a) Portfolio Optimization (4 Assets, 4q)

| Model | Repeats | GateCounts | Value | Reference |
|---|---|---|---|---|
| TwoLocal | 2 | **24** | -0.01112 | |
| | 3 | 34 | -0.01303 | |
| | 5 | 54 | -0.00645 | -0.0149 |
| RealAmplitude | 2 | 29 | 0.00053 | |
| | 3 | 39 | -0.00217 | |
| QGAS (Ours) | 1 | 31 | **-0.01347** | |

(b) Max-Cut (5nodes, 5q)

| Model | Repeats | GateCounts | Value | Reference |
|---|---|---|---|---|
| TwoLocal | 2 | **29** | -1.99694 | |
| | 3 | 38 | -1.99329 | |
| | 5 | 96 | -1.99332 | -2.0 |
| RealAmplitude | 2 | 41 | **-1.99890** | |
| | 3 | 56 | -1.99516 | |
| QGAS (Ours) | 1 | 33 | -1.99837 | |

(c) TSP (3nodes, 8q)

| Model | Repeats | GateCounts | Value | Reference |
|---|---|---|---|---|
| TwoLocal | 2 | **53** | -7317.077 | |
| | 3 | 70 | -7327.370 | |
| | 5 | 104 | -7017.373 | -7379.0 |
| RealAmplitude | 2 | 109 | -7309.703 | |
| | 3 | 154 | -6181.840 | |
| QGAS (Ours) | 1 | 58 | **-7376.639** | |

In the experiment determining molecular ground state energy, as shown in Fig. 4, we compared QGAS with the state-of-the-art ansatz architecture search framework QuantumNAS and the problem ansatz UCCSD. In noise-free conditions, QGAS achieved slightly better results than UCCSD and was comparable with QuantumNAS. We also plotted the performance of the ansatz architectures and corresponding estimated energy at different iterations during model optimization in Fig. 3. We then tested QGAS's adaptability to noise in a quantum environment. As seen in Figure 4(a), for the MNIST classification problem in a noisy environment, QGAS performed worse than QuantumNAS but significantly better than a randomly generated ansatz. We believe that enhancing QGAS's adaptability to noise should be closely linked to the combination of human feedback with the capability of GPT, which will be discussed in the next section. The primary reason for the current lack of adaptability to noise is the difficulty in successfully conveying a complex quantum noise environment to GPT. This requires further contemplation and more meticulous work to devise specific prompts.

## 5.4 OBSERVATION FROM HUMAN FEEDBACK & CAPABILITY OF GPT

Throughout the experimental process, human feedback played an indispensable role in achieving exceptional performance with GPT-4. In the most fundamental framework, we initially had to fine-tune the system with specific prompts. Without this adjustment, GPT-4 would yield choices outside of the given design space. In the experiments concerning molecular ground state energy, GPT-4 proactively analyzed and opted to improve the circuit's expressive power by increasing the circuit depth. However, when the circuit depth became too great, we observed a decrease in accuracy instead of an expected increase. Moreover, the number of epochs in the optimization process was substantial. We converted these observations into human-readable text and submitted them to GPT-4. Upon receiving our feedback, GPT-4 quickly provided several alternative solutions.

We prompt GPT-4 to generate new ansatz architecture based on itself's suggested solutions on both LiH and $H_2O$ tasks. In the $H_2O$ task, GPT-4 suggested initializing all parameters on the rotation gates to a single value, this set to a value related to molecular characteristics based on GPT-4's

analysis. However, this analysis was not entirely scientific based on our experience. Ultimately, this approach reduced the number of epochs by four while maintaining the circuit depth and performance as shown in Fig. 3(a). In the LiH task, GPT-4 deployed a $\sqrt{H}$ gate on all qubits initially, followed by an identical ansatz structure behind these $\sqrt{H}$ gates. GPT-4 referred to this form as VQE-I (VQE with Initial Guess). Remarkably, this approach reduced the number of epochs by 47 while maintaining the circuit depth and performance as shown in Fig. 3(b).

These findings underscore the essential role of human feedback in guiding GPT-4's performance in quantum circuit architecture design. By refining the system's understanding and approaches, human feedback helps to optimize the balance between circuit depth, performance, and computational efficiency. This form of interactive cooperation between humans and artificial intelligence provides a promising avenue for progress in quantum computing applications.

## 6 DISCUSSION

It is important to recognize that quantum computing is currently in its nascent stage, often referred to as the Noisy Intermediate-Scale Quantum (NISQ) era. In the NISQ regime, which is characterized by a limited number of imperfect qubits without error correction, most algorithms are variational, used to address problems in quantum chemistry, combinational optimization, and quantum machine learning. These algorithms use a measurement scheme on parameterized quantum circuits, with variations of the parameters used to optimize a cost function that estimates target observables.

- GPT can help improve these components by designing better quantum architectures. By absorbing information from classical computer architecture.
- GPT can propose architectures that are efficient in terms of experimental resources and robust to hardware noise. Additionally, the quality of the initial parameter guess heavily influences the number of iterations required for optimization.
- GPT can assist in this regard by utilizing its ability to absorb broad knowledge to make a good initial parameter guess. For example, GPT can fine-tune a parameterized circuit with initial guesses by leveraging observations from the quantum chemistry community on the properties of molecules such as electronic symmetry. This substantially reduces the quantum resources required by maximizing the usage of the prior knowledge of the interdisciplinary research community.

**Ethics.** In employing GPT for recommending quantum computing circuit structures, several ethical issues may emerge. The aspect of data privacy and security is paramount, as the data used for suggesting and optimizing quantum circuit configurations may encompass sensitive and confidential information, making it susceptible to unauthorized access. This circumstance highlights the imperative need for robust security and privacy safeguards. Moreover, decision transparency and explainability present a significant concern. The potential lack of transparency and explainability in GPT's recommendations could obfuscate the understanding and verification of the proposed decision processes and outcomes, leading to possible mistrust and skepticism. The inadvertent absorption and propagation of bias from training data by GPT can culminate in the recommended quantum computing circuit structures being unreliable, thus, human supervising is required for this kind of model.

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

# A MORE DISCUSSIONS

## A.1 NOISY SIMULATION

We conducted further experiments similar to those depicted in Figure 3 for estimating the ground state energy of the $LiH$ molecule, this time incorporating the noise model from $ibmq\_quito$ to assess the performance of QGAS in a more realistic environment. As illustrated in Figure 5, the first eight iterations employ the same circuit architecture as the experiment in Figure 3, with the sole difference being the consideration of the noise model. This adjustment led to a significant degradation in performance. Despite the ansatz architecture exhibiting enhanced performance in a noiseless environment, it failed to maintain this superiority under noisy conditions. When compared to the state-of-the-art circuit architecture search scheme QuantumNAS (Wang et al., 2022a), an estimated energy around -6.4 H is achievable. Even though a superior result is noted with the ansatz architecture generated by QGAS at the sixth iteration, it's evident that its performance is not robust in a noisy environment. GPT cannot understand the noise model, and cannot distinguish different noise models, some better prompts or framework to fine-tune GPT for noisy quantum environments is needed.

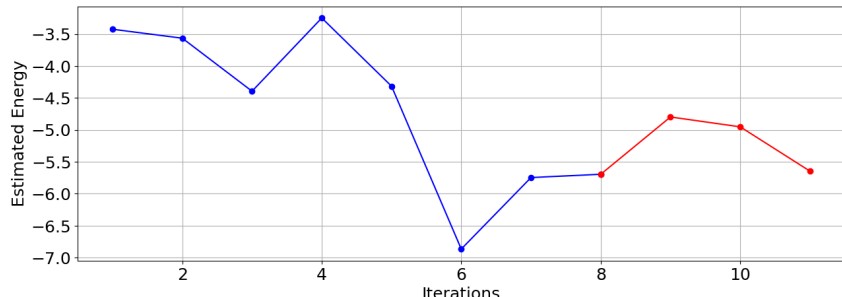

Figure 5: Experiments for the quantum chemistry molecule ground state energy tasks and evaluation of the ansatz architecture generated by QGAS. We show estimated energy for each iteration, where a lower estimated energy indicates a better ansatz. And first eight structures are adopted from Figure. 3, and the rest four are iterative recommended by GPT-4.

## A.2 WHAT ELSE LLMS CAN DO FOR QUANTUM COMPUTING?

As quantum hardware improves, fault-tolerant (FT) quantum computers may become more feasible, allowing for perfect manipulation of logical qubits. Looking forward to the future of GPT applications in this field, GPT can be used to design and optimize fault-tolerant quantum algorithms by absorbing knowledge from a variety of sources, like classical computing and quantum error correction theory. This can lead to the development of more robust and efficient quantum algorithms, bringing us closer to achieving practical quantum computing capabilities. When the era of FT quantum computing arrives, boasting formidable computational capabilities, FT quantum machines will have the potential to enable a paradigm of quantum-assisted GPT. This will facilitate large-scale training with few reliable resources, leveraging the inherent properties and immense computational power of quantum computing.

It is important to recognize that GPT's knowledge is derived from vast data models available on the internet (Hutson, 2021), and much of the public information surrounding the nascent field of quantum computing can be misleading. Moreover, a significant amount of bias exists that questions the validity of quantum computing. Due to the complexity of the knowledge required in this domain, even researchers may inadvertently propagate incorrect ideas in their publications. GPT takes all of this information into account, which may result in responses that exhibit an inadvertent bias against quantum computing. Such an effect can be particularly detrimental at this stage of quantum computing's development, as negative and harmful information could impact researchers' motivation and enthusiasm for their work. Therefore, by exploring how GPT can help quantum computing, in what form one should use GPT in the development of quantum computing, and what kind of connection

exists between quantum computing and GPT, sorting out these can reduce the risks associated with combining quantum computing and GPT.

## A.3 BEYOND REACH

GPT is not currently a general artificial intelligence; rather, it is a form of seemingly sophisticated artificial intelligence. Therefore, at least for the time being, it cannot think like a human with emotions and cannot perform any tasks that require humanity. Although GPT performs well in reading comprehension and assessment on tests such as the SAT and GRE, it is limited by the large-scale data on which it is based (Caliskan et al., 2017), i.e. it cannot observe scientific phenomena in nature by itself, it cannot monitor the entire physical environment of scientific experiments, it loses coherence in lengthy conversations, and it even contradicts its own previous conclusions (Brown et al., 2020). For quantum computing, an emerging, complex, and still-evolving field, GPT is possible to obtain incorrect information from a massive dataset. Furthermore, it cannot "think" dynamically about quantum physics and quantum information theory to design new quantum algorithms. In addition, it cannot construct a quantum computer "by hand" nor can it detect emergencies during physical experiments or interactions between quantum hardware researchers.

## A.4 LOOKING INTO THE FUTURE

Currently, the GPT model has demonstrated its capability to learn from large data and generate reasonable answers to proper questions. We expect to see it integrate better capability to reason logically for math equations. Harnessing the ability to recognize the patterns in the text and statistical associations between words and phrases to make predictions about what comes next (e.g., mathematical symbols and operations), GPT is developing a better understanding of the underlying mathematical concepts. If it were able to produce accurate logical reasoning for mathematical equations, the sprinkled knowledge across the vast quantum world could be more connected, which expedites the discovery of new quantum algorithms including more efficient error correction codes and improved mappings between fermions and qubits for Hamiltonian simulation.

For the aspect of quantum control, calibration is crucial for achieving optimal system performance. Quantum computing systems require precise calibration to ensure accurate and efficient processing of information. In this context, two typical parameters that need to be calibrated are amplitude and duration. A traditional calibration process begins by assigning a default value to the amplitude parameter. Subsequently, the duration parameter is tuned and the optimal value is determined by identifying the highest point of fidelity. Once the optimal duration is established, the amplitude is swept to further refine the system's performance. By iteratively adjusting these parameters, the quantum state can approximate the desired state. When prior knowledge of the quantum hardware is available, the noise's dominant effect can be evaluated. For instance, assume that in a 3-D plot, the amplitude and duration parameters are the X and Y axes, the fidelity of the quantum gate is the Z-axis. This configuration produces a mountainous landscape that characterizes the noise feature. If experimental data indicates that the landscape has changed, the objective is to pinpoint the optimal peak's new location. By scanning additional points, the system can be re-calibrated to account for the landscape shift caused by the hardware. Incorporating prior knowledge about the hardware and the general shape of the landscape from a few testing samples, the next generations of GPT potentially could be utilized to estimate the extent of the landscape shift caused by experimental factors. This application of GPT can potentially expedite the calibration process.

In addition to the algorithm innovations, GPT also exposes precious opportunities to validate them in an agile manner, which is critical given how fast these innovations could be. We have already witnessed the power of GPT to facilitate software development, and we believe that GPT is also capable of advancing the simulation of quantum computers. GPT can summarize the mandatory steps that occur in distinct candidate algorithms, standardize them in a universally compatible simulation framework, and provide suggestions on hardware and software resources to rapidly construct such simulation frameworks and validate the innovations. To live up to these promises, GPT needs to be further trained on quantum-computing specific datasets, and gain deeper and broader insights about how quantum computing can evolve in the foreseeable future, and we believe this is an interdisciplinary mission that brings the best of all worlds into quantum computing.

