# OpenReview forum: "Unleashing the Potential of LLMs for Quantum Computing: A Study in Quantum Architecture Design"
_ICLR.cc/2024/Conference — ICLR 2024 Conference Withdrawn Submission_

### Official Review · Reviewer_mQvQ · 2023-10-16

**Soundness:** 2 fair
**Presentation:** 3 good
**Contribution:** 2 fair
**Rating:** 3
**Confidence:** 4

**Summary:**

This paper explores the combination of Large Language Models (LLMs) with quantum computing, especially for Variational Quantum Algorithms (VQAs). A new paradigm named Quantum GPT-Guided Architecture Search (QGAS) is proposed. Specifically, GPT-4 suggests an ansatz according to the prompt, and GPT-3.5 converts it into QASM format. Experiments on Portfolio Optimization, ground state energy estimiation, combinatorial optimization, and classification demonstrate that QGAS can achieve comparable performance to SOTA methods. Furthermore, human feedback may help QGAS improve the performance.

**Strengths:**

- The paper applies LLMs to Ansatz Architecture Search for the first time, which is innovative.
- By deriving the problem hamiltonians, the paper unifies several applications such as quantum chemistry, quantum finance, and quantum optimization problems. Experiments show QGAS has comparable performance to SOTA methods.
- Some ethical issues and challenges in applying GPT to quantum computing are discussed. Also, future directions and promising applications combining LLMs and quantum computing are mentioned.

**Weaknesses:**

- Despite the novelty of applying LLMs to ansatz architechture search, the methodology is plain. How to let GPT understand the problem and the performance of the current ansatz should be elaborately designed. Also, the proposed prompts neglect the quantum noise, which weakens the performance of QGAS in noisy environments.

- The existing GPT model is not professional in quantum computing. It may generate misleading outcomes, which makes the training process unstable. Fine-tuning the LLMs in quantum computing is necessary for the robustness of QGAS.

- Since the knowledge of GPT mostly originates from existing information on the internet and the reasoning ability of it is limited, it can hardly generate some innovative ansatze leading to an outstanding result or new problem-specific ansatze.

- The experimental results are not convincing enough. An important metric in ground state energy estimation is the chemical accuracy. Whether the chemical accuracy is achieved or the distance between estimated energy and ground state energy should be noted like [1]. Besides, TwoLocal and RealAmplitude seem weak in optimization problems. Methods like [2] and [3] should be compared.

[1] Reinforcement learning for optimization of variational quantum circuit architectures[J].

[2] A quantum approximate optimization algorithm[J].

[3] From the quantum approximate optimization algorithm to a quantum alternating operator ansatz[J].

**Questions:**

1. How do you choose the six design spaces? Will the choice of design spaces influence the performance?
2. What hardware (GPUs) do you use to run the experiments? What's the total training time of an experiment?
3. According to my knowledge, UCCSD can achieve ground state energy with an extremely deep circuit. Why is the performance of UCCSD in this paper so bad?

**Details Of Ethics Concerns:**

None.

---

### Official Review · Reviewer_A6VM · 2023-10-29

**Soundness:** 1 poor
**Presentation:** 2 fair
**Contribution:** 1 poor
**Rating:** 1
**Confidence:** 4

**Summary:**

This paper presents a LLM based algorithm for Quantum Architecture Search (QAS), in which the LLM is repeatedly prompted to design a circuit then iteratively informed of its optimised performance. Using GPT4 as the LLM, the authors find that this algorithms performs well when compared to other QAS algorithms on a suite of small benchmarks.

**Strengths:**

- The paper is generally readable and the figures help with understanding (esp. Figure 1, very clear representation of the algorithm)
- Improvements in modern ML often go underutilised in QML, so it is good to see the community adopting more from the classical ML side of things

**Weaknesses:**

- The introduction would benefit from LLM citations. There is much debate surrounding the abilities of LLMs, so reinforcing the claims on their capacity with references is needed.
- The introduction should have more of the big points of what we can expect to see in the paper (beyond motivation, describe what this paper is actually doing)
- The figures would benefit from multiple runs being averaged over with some uncertainty measure. There should be error bars on the figures and uncertainty metrics for the table.
- NISQ should be cited [1]
- There doesn’t seem to be any reason to expect LLMs to excel at QAS and the idea that LLMs will have a meaningful impact on the QAS space not well justified. If the takeaway is that big models are good at QAS, then it seems like a big model should be trained for QAS.
- Given the small-scale nature of the problems and how often you see them in literature, it is possible the problems are in (or very similar) to data already in GPT4’s training set (an unfalsifiable claim, I know). This makes evaluating an apples to oranges comparison even harder.
- More rigorous analysis and problems should be evaluated. How does LLM QAS scale with depth? With width? With hamiltonian size? Etc. There are a lot of questions that go unanswered that are essential for claiming something as blackbox as GPT4 can be shown to be good at a task such as QAS.
- QAS is not a problem of great interest to the ICLR community and a paper showing GPT is decent at a (relatively) niche problem will likely not be of interest to the community as well.

References:

[1] Preskill, J. (2018). Quantum computing in the NISQ era and beyond. Quantum, 2, 79.

**Questions:**

- Would the code be available if the publication is accepted?

---

### Official Review · Reviewer_GZfc · 2023-11-01

**Soundness:** 2 fair
**Presentation:** 3 good
**Contribution:** 1 poor
**Rating:** 1
**Confidence:** 5

**Summary:**

The authors of this paper provide a high level methodology and evaluation of using Large Language Models in generating ansatz for the VQA optimization tasks. for each problem in the benchmark set, an ansatz was generated by generating a simple parametric prompt to the LLM. Then it was optimized for the target task and evaluated by comparing with other available methods. The paper is well written but its contribution is not clear.

**Strengths:**

This paper provides an alternative to soft-computing design of VQA ansatz by combining a set of existing tools. This is the interesting part because a well understood set of tools could provide a very highly valued mechanics for designing these quantum circuits.

**Weaknesses:**

The lack of explanation and proper evaluation: the methodology resides in a relatively shallow approach of how to generate ansatz with even less understanding. In general the paper goes against the understanding of methodology.

While I understand the hype from the LLMs and other generative models for providing solutions, I think blindly using these models without attempting to understand the basis of the problem is not a contribution for this venue.

**Questions:**

What is the real contribution?